# Single-Disulfide Conopeptide Czon1107, an Allosteric Antagonist of the Human α3β4 Nicotinic Acetylcholine Receptor

**DOI:** 10.3390/md20080497

**Published:** 2022-07-31

**Authors:** Yuan Ma, Qiushi Cao, Mengke Yang, Yue Gao, Shuiping Fu, Wenhao Du, David J. Adams, Tao Jiang, Han-Shen Tae, Rilei Yu

**Affiliations:** 1Key Laboratory of Marine Drugs, Chinese Ministry of Education, School of Medicine and Pharmacy, Ocean University of China, 5 Yushan Road, Qingdao 266003, China; yuanma20001008@163.com (Y.M.); cqslky@163.com (Q.C.); yangmengke@stu.ouc.edu.cn (M.Y.); 13691075226@163.com (Y.G.); fspfushuiping@163.com (S.F.); du18860530531@163.com (W.D.); jiangtao@ouc.edu.cn (T.J.); 2Illawarra Health and Medical Research Institute (IHMRI), University of Wollongong, Wollongong, NSW 2522, Australia; djadams@uow.edu.au; 3Laboratory for Marine Drugs and Bioproducts, Qingdao National Laboratory for Marine Science and Technology, Qingdao 266003, China; 4Innovation Center for Marine Drug Screening & Evaluation, Qingdao National Laboratory for Marine Science and Technology, Qingdao 266003, China

**Keywords:** conopeptide, nAChR, ACh-evoked currents, allosteric inhibitor, structure–activity relationship, molecular dynamics simulations

## Abstract

Conopeptides are peptides in the venom of marine cone snails that are used for capturing prey or as a defense against predators. A new cysteine-poor conopeptide, Czon1107, has exhibited non-competitive inhibition with an undefined allosteric mechanism in the human (h) α3β4 nicotinic acetylcholine receptors (nAChRs). In this study, the binding mode of Czon1107 to hα3β4 nAChR was investigated using molecular dynamics simulations coupled with mutagenesis studies of the peptide and electrophysiology studies on heterologous hα3β4 nAChRs. Overall, this study clarifies the structure–activity relationship of Czon1107 and hα3β4 nAChR and provides an important experimental and theoretical basis for the development of new peptide drugs.

## 1. Introduction

Acetylcholine acts on nicotinic acetylcholine receptors (nAChRs) [1], which are responsible for regulating synaptic transmission in the central and peripheral nervous systems [2,3,4,5], anti-inflammation in the cholinergic system [6], and the growth of cancer cells [7]. The nAChRs are ligand-gated ion channels in a pentameric arrangement of seventeen different subunits (α1–α10, β1–β4, δ, γ, and ε) with a central pore [8,9,10], an intracellular domain (ICD), a transmembrane domain (TMD), and an extracellular domain (ECD). The ECD of two adjacent subunits constitutes the ligand-binding site, which consists of loops A, B, and C of one subunit and the β-sheet of the complementary subunit.

Neuronal nAChRs are associated with cognitive activity, pain perception, learning and memory formation, and neuronal development in humans, thus therapeutics that act on nicotinic synaptic transmission may be useful for the treatment of pain, memory disorders, Parkinson’s disease, schizophrenia, and nicotine addiction [5,10].

The α3β4 nAChR (Figure 1) is the predominant subtype in the autonomic ganglia, adrenal medulla, and subsets of nerve cells in the medial habenula, nucleus interpeduncularis, dorsal medulla, pineal gland, and retina [11]. Furthermore, they are associated with nicotine addiction, drug abuse, and the development of lung cancer [12].

Conopeptides that include disulfide-rich conotoxins (Ctx) and non-disulfide-rich peptides are small peptides from the venom of marine *Conus* snails [13], which are used for capturing prey or as a defense against predators [13], and act on ion channels, transporters, G protein-coupled receptors, and enzymes [14,15,16,17,18]. They are categorized into superfamilies based on sequence and framework homology and pharmacological families based on their targets [19]. Several superfamilies have been characterized, including A-, O-, T-, P-, I-, S-, and M- that possess two to five disulfide cross-links [20,21,22,23,24]. 

Conopeptides have attracted attention as potential therapeutics and several have entered clinical trials such as the non-addictive analgesic ω-Ctx MVIIA [12]. In addition, α-Ctxs Vc1.1 and RgIA have been reported to exhibit analgesic properties in rat behavioral models of neuropathic pain [24,25]. Owing to their structural stability, relatively small size, and target specificity, conopeptides are regarded as ideal molecular probes for target validation and peptide drug discovery. 

Currently, most of the reported conopeptides targeting nAChRs are the widely studied A-superfamily α-Ctxs, which contain 12 to 18 amino acids including two pairs of disulfide bonds. The disulfide bridges are critical in constraining the conopeptides into a bioactive conformation and variations in cysteine connectivity also affect the pharmacological properties of conopeptides [26,27]. However, cysteine-poor conopeptide antagonists of the nAChRs are rarely studied. Therefore, this is a new field of study on the activity of conopeptides with one disulfide bond acting on nAChRs.

The *C. zonatus* Czon1107 only has a single disulfide bond and in fluorescence bioassays on heterologous hα3β4 nAChRs expressed in SH-SY5Y cells, Czon1107 non-competitively inhibits nicotine-evoked currents with a half-maximal inhibitory concentration (IC_50_) of 15.7 µM [28]. However, the allosteric mechanism of Czon1107 inhibition has not been investigated. Thus, in this study, the binding mode of Czon1107 to hα3β4 nAChR was predicted by molecular dynamics (MD) simulations and analogues of conopeptide Czon1107 were synthesized and tested on hα3β4 nAChRs heterologously expressed in *Xenopus laevis* oocytes using the two-electrode voltage clamp technique. Overall, the inhibitory mechanism of Czon1107 in hα3β4 nAChR putatively involves hydrophobic interactions between Czon1107 residue F2 and α3 residues P264 and P271, and hydrogen bonds between Czon1107 residue R3 and residues S225 and T280 of the α3 subunit.

## 2. Results and Discussion

### 2.1. Synthesis of α-conotoxin Czon1107 and Analogues

The peptides (Table 1) were synthesized by solid-phase Fmoc chemistry, as described previously [29]. As Czon1107 and its analogues contain only one disulfide bond, they can be formed directly under I_2_ oxidation. The fully oxidized product was separated and purified by reversed-phased high-performance liquid chromatography (RP-HPLC). The molecular weight and purity were confirmed by electrospray ionization–mass spectrometry (ESI–MS) and analytical HPLC (Appendix A).

### 2.2. Molecular Dynamics (MD) Simulations of hα3β4 nAChR Bound with Czon1107

CoDock is an efficient tool for peptide docking, which integrates the shape singularity, knowledge-based scoring function, and site constraint [30]. Using CoDock, we docked hα3β4 nAChR (PDB code: 6PV7) with Czon1107 (PDB code: 6KN2) several times and compared the convergence of each result. From this, we obtained the initial model (Figure 2), suggesting Czon1107 is mostly distributed in the TMD region of hα3β4 nAChR. Three docking models with the highest reproducibility (Figure 2C–E) were selected for the 200 ns molecular dynamics simulation (MD) (Appendix A).

As the three suggested Czon1107 binding sites are located at the lower half of the receptor, only the hα3β4 nAChR TMD and ICD models were simulated. The MD simulations proposed that the interaction site is located at the junction of the α3 subunit’s ECD and TMD (Figure 2C) as it had the most favorable physicochemical environment such as hydrogen bonds and hydrophobic interactions. 

### 2.3. Activity of Czon1107 Analogues at hα3β4 nAChRs

Mutational analysis is widely used to study the relationship between the sequence and structure of peptides [28,31]. In order to identify the residues involved in the interaction between hα3β4 nAChR and conopeptide Czon1107, an alanine (Ala)-scan mutation was carried out on Czon1107. The activity of the synthesized Czon1107 Ala analogues at 10 µM (close to the reported IC_50_ of Czon1107 at hα3β4 nAChR [28]) was determined on ACh-evoked currents at hα3β4 nAChRs (Figure 3 and Appendix A). All peptides inhibited hα3β4-mediated ACh currents in a reversible manner. Three of the Ala analogues (G1A, P8A, and F9A) exhibited similar activity to Czon1107, whereas each Ala substitution of residues from Phe at position 2 to Ser at position 4 (F2A–S4A), significantly reduced the activity of the peptide (10–20% inhibition, *p* < 0.05, *n* = 5–6). 

Based on the results of MD, Czon1107 is putatively embedded in a binding pocket (Figure 4A) on the α3 subunit. In the binding model (Figure 4B,C), residue F2 forms hydrophobic interactions with the α3 subunit residues P264 and P271, whereas the R3 residue is relatively crucial as it forms hydrogen bonds with α3 S225 and T280 (Figure 4E). Thus, it is speculated that Ala substitution at both Czon1107 residues abolishes the interactions as reflected in the significant loss of inhibitory activity on hα3β4 ACh-evoked currents. Additionally, the hydroxyl group of S4 played a role in sustaining the stability of the peptide by forming intramolecular hydrogen bonds with the carbonyl group of F2. Thus, the reduced activity of the S4A analogue could be attributed to peptide stability perturbation. In terms of spatial position, Czon1107 P8 is externally oriented toward the α3 subunit, therefore it lacks a suitable interaction force with the surroundings. Residue G1 had no obvious interactions with the α3 subunit, thus, alanine replacement had no effect on the activity of the peptide. F9 and L468 formed a hydrophobic interaction and Van der Waals’ force (VDW), and removal of the aromatic side chain of F9 resulted in a slight potency decrease for the F9A mutant. 

To further investigate the effects of charge on electrophysiological activity and potentially optimize the activity, we performed an arginine (Arg)-scan mutagenesis on Czon1107. The replacement of residues at positions 1 and 7 with the positively charged Arg slightly enhanced the potency of Czon1107 in inhibiting the hα3β4 nAChRs (Figure 3). Both the G1R and P7R analogues inhibited ~50% of the ACh-evoked current amplitude compared to ~30% inhibition by Czon1107 (*p* < 0.0001, *n* = 6). The improved activity of Czon1107[G1R] could be explained by the formation of hydrogen bonds with the carbonyl and hydroxyl groups of T262 (Figure 4D). Furthermore, swapping P7 with Arg residue putatively formed hydrogen bonds (Figure 4F) with the hydroxyl groups of T214 and carbonyl groups of P210. 

In contrast, Arg substitution of residues F2, S4, and F9 reduced the inhibitory activity of the peptide at hα3β4 nAChRs. In comparison to Czon1107, all three analogues inhibited the ACh-evoked current amplitude by 5–20% (*p* < 0.05, *n* = 7–9). The hydrophobic interactions formed between F2 and P264 and P271 were removed by the introduction of arginine at this position, resulting in a significant decrease in activity for Czon1107[F2R]. Similarly, the F9R mutation resulted in a decrease in inhibitory activity due to the removal of the hydrophobic interactions and the VDW between F9 and L468. In addition, as the S4 residue is embedded in the binding pocket, S4R mutation could introduce steric hindrance, which could be responsible for the significant decrease in activity for Czon1107[S4R] in comparison to the wild-type peptide. No significant change in activity at hα3β4 nAChRs was observed for Czon1107[P5R] and Czon1107[P8R].

In addition to the hα3β4 nAChRs, Czon1107 inhibited nicotine-evoked currents at hα7 nAChRs (IC_50_ of 77.2 µM) [28]. A comparison between hα3β4 and hα7 putatively suggested that Czon1107 residue F2 likely forms a hydrophobic interaction with the hydrophobic P264 residue of hα3β4 nAChR compared to the corresponding hydrophilic and charged D266 residue of hα7 (Appendix A). To some extent, our model also explains the selectivity of the spatial position between hα3β4 and hα7 nAChRs.

## 3. Materials and Methods 

### 3.1. Synthesis of α-Conotoxin Czon1107 and Its Analogues

Peptides were synthesized using the standard solid-phase peptide synthesis (SPPS) as reported previously [29]. After saturating with a mixed solution of dimethylformamide (DMF): dichloromethane (DCM) (1:1) for 4 h, 20% of piperidine solution was used in rink amide resin (loading amount 0.631 mmol/g) in order to remove the Fmoc protecting group (30 min). Amino acids (4.0 equivalent), (O-^1^H-6-chlorobenzotriazole-1-yl)-1,1,3,3-tetramethylu-ronium hexafluoro-phosphate (HCTU; 4.0 equivalent) and N, N-diisopropylethylamine (DIPEA; 8.0 equivalent) were added in DMF for 1 h at room temperature and washed three times with DCM and DMF. After the completion of the chemical reaction, 20% piperidine solution (30 min) was used to remove the last Fmoc protective group and washed three times with DCM and DMF. Addition of 10 mL trifluoroacetic acid (TFA): triisopropylsilane: water (9:0.5:0.5) were reacted at a constant temperature on a shaker for 3 h to remove the peptide from the resin. After washing the resin with DCM three times, the filtrate was combined. After removal of excess TFA, the peptide was then precipitated in cold diethyl ether, washed three times with diethyl ether, and dissolved in an H_2_O/acetonitrile (1:1) mixture at a peptide concentration of 0.1 mg/L. Then, the reaction was purified by reversed-phased high-performance liquid chromatography (RP-HPLC) (Tabitha^TM^ WD-C18 column 20 × 250 mm, 10 μm), using a gradient of buffer A (90% H_2_O, 10% acetonitrile, 0.05% TFA) and B (90% acetonitrile, 10% H_2_O, 0.05% TFA) of 100% to 40% for 1 h with a flow rate of 6 mL/ min. Subsequently, the peptide was oxidized by adding 5 mg/mL iodoacetonitrile under a closed environment at 28 °C for about 2 h. After the reaction was finished, an ascorbic acid buffer was added to quench the reaction. The oxidized product was purified by RP-HPLC, and the separation conditions were the same as above. This was followed by concentration under vacuum and lyophilization of the peptide as a white powdery solid; the product was stored at −20 °C.

### 3.2. Xenopus laevis Oocyte Preparation and Microinjection

All procedures were approved by the University of Wollongong Animal Ethics Committees (project number AE2003). Female *X. laevis* were sourced from Nasco (Fort Atkinson, WI, USA) and a maximum of four frogs were kept in a 15 L aquarium at 20–26°C with a 12 h light/dark cycle. Oocytes (Stage V–VI Dumont’s classification; 1200–1300 μm in diameter) were removed from *X. laevis* by surgical laparotomy and defolliculated with 1.5 mg/mL collagenase Type II (Worthington Biochemical Corp., Lakewood, NJ, USA). Oocytes were obtained from three frogs. Defolliculation was done at room temperature (21–24 °C) for 1–2 h in OR-2 solution containing (in mmol/L) 82.5 NaCl, 2 KCl, 1 MgCl_2_, 5 HEPES, pH 7.4. Plasmid pT7TS constructs of hα3 and hβ4 were linearized with XbaI (NEB, Ipswich, MA, USA) for in vitro T7 mMessage mMachine®-cRNA transcription (AMBION, Foster City, CA, USA). Oocytes were injected with 5 ng of hα3β4 cRNAs at α3 to β4 ratio of 1:1 (concentration confirmed spectrophotometrically and by gel electrophoresis) using glass pipettes as described previously [32]. Oocytes were incubated at 18 °C in sterile ND96 solution composed of (in mM) 96 NaCl, 2 KCl, 1 CaCl_2_, 1 MgCl_2_, and 5 HEPES at pH 7.4, supplemented with 5% fetal bovine serum (FBS), 50 mg/L gentamicin (GIBCO, Grand Island, NY, USA) and 10,000 U/mL penicillin-streptomycin (GIBCO).

### 3.3. Two-Electrode Voltage Clamp Recording of Oocytes and Data Analysis

Two-electrode voltage clamp recordings were carried out on *X. laevis* oocytes expressing hα3β4 nAChRs. Voltage clamp recordings were performed using a GeneClamp 500B amplifier and pClamp9 software interface (Molecular Devices, Sunnyvale, CA, USA) at a holding potential of −80 mV at room temperature. Oocytes were impaled with two microelectrodes filled with 3 mol/L KCl. Voltage recording and current injection electrodes were both made of GC150T-7.5 borosilicate cate glass (Harvard Apparatus, Holliston, MA, USA) and had a resistance of 0.3–1 MΩ when filled with 3 mol/L KCl. Oocytes were continuously perfused with ND96 solution (2 mL/min). Three ACh applications were then performed with 300 µM ACh solution (half-maximal effective concentration at hα3β4 nAChR) [32]. After perfusion was stopped, oocytes were incubated with the peptide for 5 mins and then with flowing ND96 solution, ACh was added to peptide co-application. There was a 3 min washout time between perfusions. All peptides were tested at 10 µM, close to the reported IC_50_ of wild-type Czon1107 at hα3β4 nAChRs [28]. The peptide solutions were prepared in ND96 + 0.1% FBS. Oocytes were incubated with 0.1% FBS to ensure that the pressure of the FBS and perfusion system had no effect on the nAChRs. Peptides were tested on the same oocyte whenever possible. Peak ACh-evoked currents before and after peptide incubation were quantitated using Clampfit 10.7 software (Molecular Devices, Sunnyvale, CA, USA) and relative current amplitudes (I_ACh+peptide_ /I_ACh_ ) were applied to assess peptide activity at hα3β4 nAChRs. The activity of the peptides was compared using an unpaired Student’s *t*-test (GraphPad Prism 9 Software, La Jolla, CA, USA), and it was deemed statistically significant with values of *p* < 0.05.

### 3.4. Docking and Molecular Dynamics (MD) Simulations

The crystal structure of the protein was retrieved from the Protein Data Bank (PDB). Czon1107-WT was docked to the hα3β4 nAChR in the CoDockPP web server (http://codockpp.schanglab.org.cn/, accessed on 17 November 2021). The produced top 10 docked conformations of the Czon1107-WT were selected for further analysis and those with the optimum binding modes were used for subsequent MD simulations to further optimize the structure.

MD simulations were carried out in AMBER 20 software using ff19SB force field [33]. The hα3β4 nAChR bound with Czon1107 was inserted into a bilayer containing a 2:2:1 mixture of POPC (1-palmitoyl-2-oleoyl-sn-glycero-3-phosphocholine): POPE(1-palmitoyl-2-oleoyl-sn-glycero-3-phosphoethanolamine): cholesterol, similar to Grossfield et al. [34], with a dimension of 125 × 125 × 129 Å and the system solvated with OPC water molecules. Then, Na^+^ was added to the whole system to make it appear electrically neutral with an overall concentration of 0.15 M in CHARMM-GUI (http://www.charmm-gui.org, accessed on 18 January 2022) [35] and energy optimization of the whole system was carried out. The temperature of the system was gradually increased to 310 K and was equilibrated for 500 Ps in NVT and NPT ensembles, respectively, with the protein and lipids restrained with 10 kcal/mol/Å2 force. For the second heating phase, an anisotropic Berendsen weak-coupling barostat was used to equilibrate the pressure in addition to the use of the Langevin thermostat to equilibrate the temperature. Then, the position restraints on the membrane were withdrawn and the system was simulated for 20 ns in NPT. The position restraints on the protein were then gradually withdrawn in 10 steps of 5 ns MD simulations. Afterward, an unrestrained production run was performed. In the production run, the temperature was controlled using the Langevin thermostat and pressure was controlled using the anisotropic Berendsen barostat. The software VMD [36] (http://www.ks.uiuc.edu/, accessed on 26 January 2022) was used to analyze the motion trajectory after MD simulations and calculate the RMSD value of the resulting conformation.

## 4. Conclusions

Conopeptides such as AuIB [37], RegIIA [38] and their analogues antagonize α3β4 nAChR in a competitive manner. Some conopeptides, including the globular disulfide isomer of AuIB [26], GeXIVA [39], and ImII [40], which have multiple disulfide bonds and complex synthesis pathways, have been suggested as non-competitive nAChR antagonists. Uniquely, Czon1107, a single disulfide bond conopeptide, also inhibits the α3β4 nAChR in a non-competitive manner. 

This study gives insight into the structure–activity relationship of Czon1107 and hα3β4 nAChR. Through molecular docking and MD simulations, the favorable binding site for Czon1107 is putatively located at the α3 subunit. The MD simulations of the interactions also suggest that Czon1107 residue F2 forms hydrophobic interactions with residues P264 and P271 of the α3 subunit. Furthermore, Czon1107 residue R3 interacts with residues S225 and T280 of the receptor via hydrogen bonds. These findings provide an experimental and theoretical basis for the structure optimization and mechanisms of action of Czon1107 and its analogues. Based on the current experimental results, multi-site mutation of Czon1107 and α3 subunit is a promising research direction for elucidating a potent conopeptide at the α3β4 nAChR.

## Figures and Tables

**Figure 1 marinedrugs-20-00497-f001:**
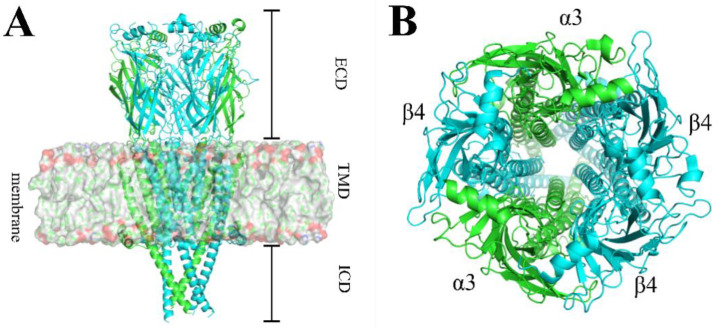
Structure of the human (h)α3β4 nAChR (PDB code: 6PV7). (**A**). The human (h) α3β4 nAChR structure comprises three regions, including a ligand-binding extracellular domain (ECD), a transmembrane domain (TMD), and an intracellular domain (ICD). (**B**). Top view of the hα3β4 nAChRs. The α3 subunit is depicted in green and the β4 subunit is depicted in cyan.

**Figure 2 marinedrugs-20-00497-f002:**
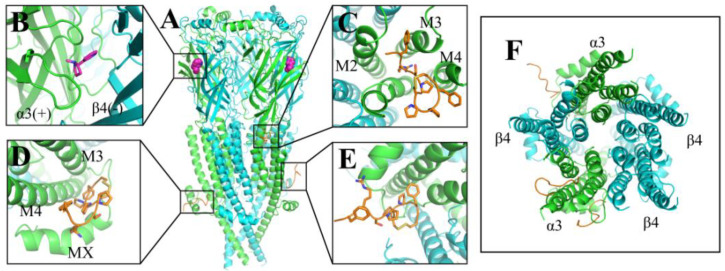
Structure of the human (h)α3β4 nAChR (PDB ID: 6PV7) (**A**). Side view of hα3β4 nAChR docked with Czon1107. (**B**). A magnified view of hα3β4 nAChR bound with nicotine (magenta). (**C**–**E**). Postulated binding sites of Czon1107 at hα3β4 nAChR. The α3 and β4 subunits are colored green and cyan, respectively, and conopeptide Czon1107 is orange. M2, M3, M4, and MX represent the TMD α-helices. (**F**). Top view of the hα3β4 nAChR TMD docked with Czon1107.

**Figure 3 marinedrugs-20-00497-f003:**
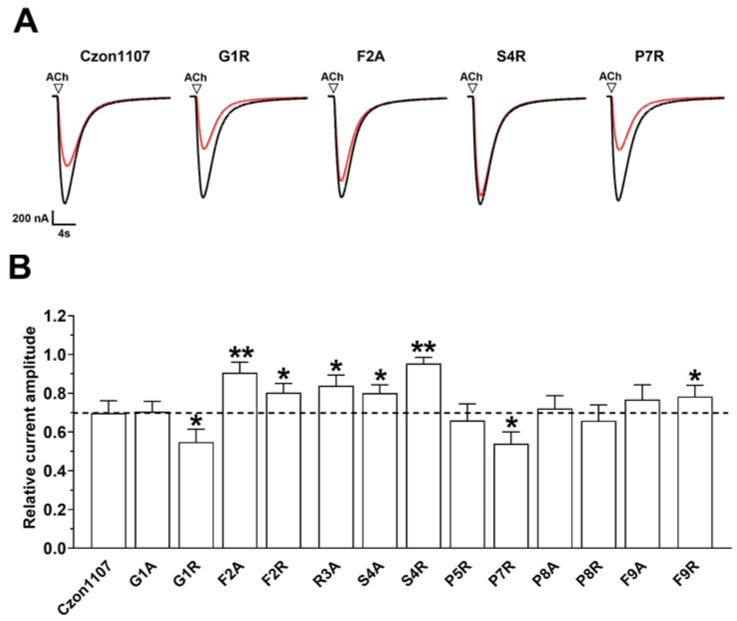
The inhibitory activity of Czon1107 and its analogues (10 μM) on ACh-evoked peak current amplitude mediated by hα3β4 nAChRs. (**A**). Representative superimposed ACh (300 µM)-evoked currents mediated by hα3β4 nAChRs, obtained in the absence (control, black trace) and presence of 10 μM Czon1107 and its analogues (red lines). (**B**). Bar graphs of relative inhibition by Czon1107 analogues (10 μM) of ACh-evoked current amplitude mediated by hα3β4 nAChRs (mean ± SD, *n* = 5–8) (* *p* < 0.05, ** *p* < 0.0001). The dashed line indicates relative inhibition of wild-type Czon1107.

**Figure 4 marinedrugs-20-00497-f004:**
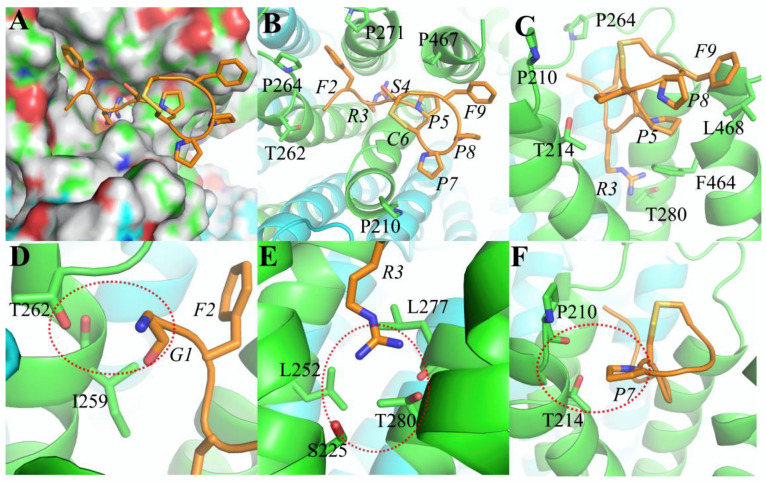
Binding modes of Czon1107 at hα3β4 nAChR. (**A**). Czon1107 is embedded in the α3 subunit binding pocket via hydrogen bonds. (**B**,**C**). Top and side views of Czon1107 at the α3 subunit, respectively. Various amino acid residues of Czon1107 form interactions with amino acids of hα3β4 nAChR. The α3(+) interface is shown in green, β4(−) in cyan and Czon1107 in orange. (**D**). The substituted arginine residue of Czon1107[G1R] putatively forms H-bonds (red dashed line) with the carbonyl and hydroxyl groups of hα3β4 threonine (T262). (**E**). The arginine residue (R3) of Czon1107 forms H-bonds (red dashed line) with hα3β4 threonine (T280) and serine (S225). (**F**). The substituted arginine residue of Czon1107[P7R] forms H-bonds (red dashed line) possibly with the hydroxyl groups of hα3β4 threonine (T214) and carbonyl groups of proline (P210).

**Table 1 marinedrugs-20-00497-t001:** Czon1107 and its analogues.

No.	Name	Sequence
1	Czon1107	GFRSPCPPFC#
2	Czon1107-G1A	AFRSPCPPFC#
3	Czon1107-F2A	GARSPCPPFC#
4	Czon1107-R3A	GFASPCPPFC#
5	Czon1107-S4A	GFRAPCPPFC#
6	Czon1107-P8A	GFRSPCPAFC#
7	Czon1107-F9A	GFRSPCPPAC#
8	Czon1107-G1R	RFRSPCPPFC#
9	Czon1107-F2R	GRRSPCPPFC#
10	Czon1107-S4R	GFRRPCPPFC#
11	Czon1107-P5R	GFRSRCPPFC#
12	Czon1107-P7R	GFRSPCRPFC#
13	Czon1107-P8R	GFRSPCPRFC#
14	Czon1107-F9R	GFRSPCPPRC#

#-indicates C-terminal amidation

## Data Availability

Not applicable.

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
