# Peer review of "Single-Disulfide Conopeptide Czon1107, an Allosteric Antagonist of the Human α3β4 Nicotinic Acetylcholine Receptor"

_marinedrugs, 2022, doi:10.3390/md20080497_

Round 1
Reviewer 1 Report
The article "α-Conotoxin Czon1107, an Allosteric Antagonist of the Human α3β4 Nicotinic Acetylcholine Receptor" by Ma et al. describes the pharmacological interaction of the conotoxin Czon1107 and the human α3β4 nAChR using electrophysiology, variants of the toxin and a docking model. It is concluded that Czon1107 binds to the intracellular part of the α3 subunit and the putative interacting amino acids of both the toxin and the receptor are proposed.
11. Why did the authors choose to work on the α3β4 receptor, which is probably not the most relevant for characterizing the activity of an α-conotoxin? Indeed, as noted in the text, this receptor is expressed in the PNS at the level of ganglia of the vegetative nervous system and chromaffin cells of the adrenal medulla. Why didn't they consider working with the muscle nAChR? What is the rationale for working with the α3β4 nAChR ?
22. Figure 2: A side view is proposed to show the potential interaction of the α3β4 receptor with Czon1107. To complement this, it would be interesting to have a top view of the membrane (like in Fig. 1B), extracellular side. This would clearly show the interaction sites of the toxin and nicotine, relative to each subunit.
33. In electrophysiology, if the ratio of α3β4 subunits is 1:1, it can be assumed to be expressed in two stoichiometric forms: α32β43 and α33β42. Yet the authors based their analysis on the α32β43 form. How can the levels of cell current inhibition by the toxin be interpreted if both forms of receptor co-exist?
Figure 3 is incomplete. It only offers the levels of inhibition of Czon1107 toxin and its variants on ACh-induced current (300 µM) on α3β4 receptor-expressing oocytes. The concentration of ACh is estimated to correspond to the EC50 for the α3β4 receptor but the dose-response curve is not performed in this study (simply proposed as a reference). This curve should be shown in this figure. The authors should also add to this figure representative current traces of each toxin (Czon1107 and its analogues), including a control current with ACh, and the current with ACh + Czon1107 or its analogues.
44. The results are commented but not discussed. Clearly a real discussion is missing in this work, which would allow to confront the proposed data to the already known data on the topic of α-conotoxins and nAChRs.
Minor
L33: a transmembrane domain (TMD)
Author Response
- Why did the authors choose to work on the α3β4 receptor, which is probably not the most relevant for characterizing the activity of an α-conotoxin? Indeed, as noted in the text, this receptor is expressed in the PNS at the level of ganglia of the vegetative nervous system and chromaffin cells of the adrenal medulla. Why didn't they consider working with the muscle nAChR? What is the rationale for working with the α3β4 nAChR ?
Response:
The original work by Mohan et al. (2020), assessed the biological activity of Czon1107 at major ‘conotoxin’ targets including the oxytocin (hOTR), vasopressin (hV1aR and hV2aR), NMDA (NR1-1a/2A), muscarinic (M1 and M3) and nicotinic acetylcholine receptors (α7 and α3β4 nAChRs), and voltage-gated calcium (Cav) and sodium channels (Nav). Only the nAChRs were inhibited by Czon1107 and subsequent experiments established its selectivity for α3β4 nAChRs. In addition, the mechanism of allosteric antagonism by Czon1107 at α3β4 nAChRs was investigated. Thus, we designed single amino acid analogues of Czon1107 and determined the electrophysiological activity on α3β4 nAChRs to investigate this mechanism.
- Figure 2: A side view is proposed to show the potential interaction of the α3β4 receptor with Czon1107. To complement this, it would be interesting to have a top view of the membrane (like in Fig. 1B), extracellular side. This would clearly show the interaction sites of the toxin and nicotine, relative to each subunit.
Response:
A top view of hα3β4 nAChR docked with Czon1107 is now added as Figure 2F.
- In electrophysiology, if the ratio of α3β4 subunits is 1:1, it can be assumed to be expressed in two stoichiometric forms: α32β43 and α33β42. Yet the authors based their analysis on the α32β43 form. How can the levels of cell current inhibition by the toxin be interpreted if both forms of receptor co-exist? Figure 3 is incomplete. It only offers the levels of inhibition of Czon1107 toxin and its variants on ACh-induced current (300 µM) on α3β4 receptor-expressing oocytes. The concentration of ACh is estimated to correspond to the EC50 for the α3β4 receptor but the dose-response curve is not performed in this study (simply proposed as a reference). This curve should be shown in this figure. The authors should also add to this figure representative current traces of each toxin (Czon1107 and its analogues), including a control current with ACh, and the current with ACh + Czon1107 or its analogues.
Response:
Due to the similarity of the α32β43 and α33β42 forms, we can analyze the mechanism of action based on the MD results of α32β43.
The ACh EC50 for hα3β4 nAChR used in this study is based on Cuny et al., 2016 (DOI: 10.1074/jbc.M116.730804; ref. no. 34). We have added the citation for this paper in the text and references. We have also included representative traces for Czon1107 and four of its analogues (G1R, F2A, S4R and P7R) as Figure 3A.
- The results are commented but not discussed. Clearly a real discussion is missing in this work, which would allow to confront the proposed data to the already known data on the topic of α-conotoxins and nAChRs.
Response:
We have revised the Conclusion section to “Conopeptides such as AuIB [39], RegIIA [40] and their analogues antagonize α3β4 nAChR in a competitive manner. Some conopeptides including the globular disulfide isomer of AuIB [28], GeXIVA [41] and ImII [42], which have multiple disulfide bonds and complex synthesis pathways, have been suggested as non-competitive nAChR antagonists. Uniquely, Czon1107, a single disulfide bond conopeptide, also inhibits the α3β4 nAChR in a non-competitive manner.
This study gives insight into the structure–activity relationship of Czon1107 with hα3β4 nAChR. Through molecular docking and MD simulations, the favorable binding site for Czon1107 is putatively located at the α3 subunit .MD simulations of the interactions also suggest Czon1107 residue F2 forms hydrophobic interactions with residues P264 and P271 of the α3 subunit. Furthermore, Czon1107 residue R3 interacts with residues S225 and T280 of the receptor via hydrogen bonds. These findings provide an experimental and theoretical basis for the structural optimization and mechanism of action of Czon1107 and its analogues. Based on the current experimental results, multi-site mutation is a promising research idea for finding a potent conopeptide at the α3β4 nAChR.”
Minor
L33: a transmembrane domain (TMD)
Response:
Revised as suggested.
Reviewer 2 Report
A study by Yuan Ma et al. is a continuation of the characterization of a new peptide (Czon1107) isolated earlier from the venom of Conus zonatus [Mohan et al JBC 2020]. In that work, NMR structure of Czon1107 was determined and a primary analysis was carried out to search for possible targets of its action among various receptors and ion channels that revealed affinity towards two nAChR. This work represents the second attempt at structural and functional characterization of Czon1107, for which the authors carried out its alanine- and arginine-scanning mutagenesis and obtained a series of new analogues that were investigated for inhibitory ability against the human α3β4 receptor. A significant part of the work was computer modeling with molecular dynamics in order to identify and characterize the allosteric binding site of Czon1107 on this receptor. Unfortunately, the new synthetic analogues obtained in this work did not add much in scientific terms to the reliable identification and characterization of the binding site of this peptide at hα3β4 nAChR, since the main results obtained are mostly hypothetical.
The main questions that arose when reading this manuscript are the following:
1. The attribution of Czon1107 peptide to α-conotoxins cannot be considered reasoned. Its single-disulfide cysteine framework does not correspond to α-conotoxins. Belonging to the A-superfamily has not been confirmed because there is no analysis of gene sequence from transcriptomic data. Finally, the revealed affinity for the two nAChR subtypes is quite low (> 10 μM). A much more reasonable conclusion seems to be the discovery of a new type of cysteine-poor conopeptides with an unknown target of action, which showed a low affinity for distinct nAChR subtypes, than the attribution of Czon1107 to α-conotoxins. In this regard, it is worth noting that there are already numerous examples of the action of known ligands on a "second" target with an efficiency significantly higher than the affinity of Czon1107 to nAChR (for example, the action of potassium channel blockers from scorpion venom on the same nAChRs). It is important that the discoverers of Czon1107 themselves (see [Mohan et al JBC 2020]) carefully avoided calling it α-conotoxin (with one exception "for the purpose of the following discussion"), reasonably designating it a conopeptide. Taking into account all of the above, I strongly recommend at the moment to transfer Czon1107 from α-conotoxin to conopeptide.
2. The synthesized set of new single-mutated alanine and arginine analogues of Czon1107, unfortunately, did not lead to the discovery of a compound with an essentially changed activity to the receptor, although some of them showed a significantly reduced/improved affinity at the level of 10-20%. Taking into account the initially very low affinity of the wild-type compound (15 μM), it seems rather problematic to draw any conclusions on the structural organization of the binding site (especially allosteric one, little characterized earlier). It is clear that an additional series of mutants is required here, including compounds with multiple mutations, but this is the future work. According to the data presented, the work clearly lacks experimental studies. To smooth out this impression, I recommend supplementing Figure 3 with the raw traces of Czon1107 and some its analogues from electrophysiological measurements.
3. The computer part of the study offers us the allosteric binding site of Czon1107 on hα3β4 nAChR and even indicates possible atomic bonds between the amino acid residues of the α3 subunit and the peptide. That would be a remarkable scientific advance. However, given the low affinity of the peptide itself and not very noticeable changes in the affinity of its analogues, the presence of this allosteric site requires confirmation by appropriate mutagenesis of the receptor, but for now it remains very hypothetical, that implies a softening of the formulation of the final conclusions. There is also some confusion in the manuscript with the numbering of amino acid residues in the sequence of the α3 subunit. The N-terminal ECDs of all nAChR subunits consist of approximately 200-220 amino acid residues, so the amino acid numbers in TMD should start from this numbers. What are these residues - P(34, 88, 85), T(38, 86, 104), L(76, 101), S49, I83 –- attributed to TMD in Figure 4? Where is the numbering from? If you think this numbering is correct, then give in one of the figures the sequence of α3 subunits with this numbering.
Minor remarks:
1. Line 32 – not sigma-, but gamma-subunit.
2. Line 34 – ring (??) A, B,… Do you mean loops A, B,…?
3. Lines 50 and 67 – reference [14] is out of place here.
4. Lines 76-77 – please, specify the amino acid residues of which receptor subunit are indicated.
5. Line 112 – reference [34] is identical to reference [30]. Please, correct the list of references.
6. Lines 228-229 – For this kind of electrophysiological research, it does not seem very legitimate to use the Student's test in statistical analysis. It would be more correct to use ANOVA and Post-hoc test.

Author Response
- The attribution of Czon1107 peptide to α-conotoxins cannot be considered reasoned. Its single-disulfide cysteine framework does not correspond to α-conotoxins. Belonging to the A-superfamily has not been confirmed because there is no analysis of gene sequence from transcriptomic data. Finally, the revealed affinity for the two nAChR subtypes is quite low (> 10 μM). A much more reasonable conclusion seems to be the discovery of a new type of cysteine-poor conopeptides with an unknown target of action, which showed a low affinity for distinct nAChR subtypes, than the attribution of Czon1107 to α-conotoxins. In this regard, it is worth noting that there are already numerous examples of the action of known ligands on a "second" target with an efficiency significantly higher than the affinity of Czon1107 to nAChR (for example, the action of potassium channel blockers from scorpion venom on the same nAChRs). It is important that the discoverers of Czon1107 themselves (see [Mohan et al JBC 2020]) carefully avoided calling it α-conotoxin (with one exception "for the purpose of the following discussion"), reasonably designating it a conopeptide. Taking into account all of the above, I strongly recommend at the moment to transfer Czon1107 from α-conotoxin to conopeptide.
Response:
We have now referred to Czon1107 as a conopeptide.
- The synthesized set of new single-mutated alanine and arginine analogues of Czon1107, unfortunately, did not lead to the discovery of a compound with an essentially changed activity to the receptor, although some of them showed a significantly reduced/improved affinity at the level of 10-20%. Taking into account the initially very low affinity of the wild-type compound (15 μM), it seems rather problematic to draw any conclusions on the structural organization of the binding site (especially allosteric one, little characterized earlier). It is clear that an additional series of mutants is required here, including compounds with multiple mutations, but this is the future work. According to the data presented, the work clearly lacks experimental studies. To smooth out this impression, I recommend supplementing Figure 3 with the raw traces of Czon1107 and some its analogues from electrophysiological measurements.
Response:
Although the Czon1107 analogues showed minimal changes in activity, we can determine whether the side chains of related amino acids play a role in the interactions based on these results. We also propose to carry out multiple site mutations based on the current results in future work to find peptides with enhanced activity. We have also included representative traces for Czon1107 and four of its analogues (G1R, F2A, S4R and P7R) as Figure 3A.
- The computer part of the study offers us the allosteric binding site of Czon1107 on hα3β4 nAChR and even indicates possible atomic bonds between the amino acid residues of the α3 subunit and the peptide. That would be a remarkable scientific advance. However, given the low affinity of the peptide itself and not very noticeable changes in the affinity of its analogues, the presence of this allosteric site requires confirmation by appropriate mutagenesis of the receptor, but for now it remains very hypothetical, that implies a softening of the formulation of the final conclusions. There is also some confusion in the manuscript with the numbering of amino acid residues in the sequence of the α3 subunit. The N-terminal ECDs of all nAChR subunits consist of approximately 200-220 amino acid residues, so the amino acid numbers in TMD should start from this numbers. What are these residues - P(34, 88, 85), T(38, 86, 104), L(76, 101), S49, I83 – attributed to TMD in Figure 4? Where is the numbering from? If you think this numbering is correct, then give in one of the figures the sequence of α3 subunits with this numbering.
Response:
Although we did not perform mutagenesis of the receptor, based on our model, we could explain the selectivity of Czon1107 for hα3β4 versus hα7. Residue F2 is more likely to form hydrophobic interaction with residue P264 of hα3β4, compared to the corresponding hydrophilic and charged D266 of hα7. We have revised Section 2.3.
We have numbered the sequences of α3β4 nAChR as originally reported.
Minor remarks:
- Line 32 – not sigma-, but gamma-subunit.
Response:
Revised as suggested
- Line 34 – ring (??) A, B,… Do you mean loops A, B,…?
Response:
Revised as suggested
- Lines 50 and 67 – reference [14] is out of place here.
Response:
Both errors have been corrected. The correct reference for line 50 is Terlau and Olivera, 2004 [ref. no. 14] and for line 67, the correct reference is Mohan et al., 2020 [ref. no. 30].
- Lines 76-77 – please, specify the amino acid residues of which receptor subunit are indicated.
Response:
Czon1107 interacts with the α3 subunit of α3β4 nAChR. We have revised the lines 76-77 to “Overall, the inhibitory mechanism of Czon1107 at hα3β4 nAChR putatively involves hydrophobic interactions between Czon1107 residue F2 with α3 residues P264 and P271, and hydrogen bonds between Czon1107 residue R3 and residues S225 and T280 of the α3 subunit.”
- Line 112 – reference [34] is identical to reference [30]. Please, correct the list of references.
Response:
Revised as suggested
- Lines 228-229 – For this kind of electrophysiological research, it does not seem very legitimate to use the Student's test in statistical analysis. It would be more correct to use ANOVA and Post-hoc test.
Response:
The Student’s T-test is used appropriately to compare between the means of two groups (e.g. wild-type versus mutant peptide).
Reviewer 3 Report
Czon1107 is a newly discovered Conotoxin identified from the venom of marine Conus snails. This peptide has a single disulfide bond and can have broad therapeutic potential for treating pain and memory disorders including Parkinson’s disease and nicotine addiction. Although it is known that Czon1107 can inhibit human α3β4 (neuronal nicotinic acetylcholine receptors) however the residues that are involved and the binding mode of inhibition is not thoroughly investigated. In this manuscript, the authors carried out molecular dynamics simulations to dock hα3β4 with Czon1107 to predict hot-spot residues and the types of molecular interaction that were responsible for the activity of Czon1107. The authors then synthesized several Alanine/Arginine substituted analogs of Czon1107 using solid-phase peptide synthesis and tested its activity in a biological assay. The results reported here clearly point out which all residues and interaction plays an important role in the inhibitory activity of Czon1107, an important step in developing new peptide drugs to manage pain and memory disorders. Experiments have been carried out well and the authors have an excellent data-driven explanation. Overall this is a very well-written manuscript and it absolutely fits the scope of Marine Drugs. The publication is recommended.
Author Response
Reviewer 1
- Why did the authors choose to work on the α3β4 receptor, which is probably not the most relevant for characterizing the activity of an α-conotoxin? Indeed, as noted in the text, this receptor is expressed in the PNS at the level of ganglia of the vegetative nervous system and chromaffin cells of the adrenal medulla. Why didn't they consider working with the muscle nAChR? What is the rationale for working with the α3β4 nAChR ?
Response:
The original work by Mohan et al. (2020), assessed the biological activity of Czon1107 at major ‘conotoxin’ targets including the oxytocin (hOTR), vasopressin (hV1aR and hV2aR), NMDA (NR1-1a/2A), muscarinic (M1 and M3) and nicotinic acetylcholine receptors (α7 and α3β4 nAChRs), and voltage-gated calcium (Cav) and sodium channels (Nav). Only the nAChRs were inhibited by Czon1107 and subsequent experiments established its selectivity for α3β4 nAChRs. In addition, the mechanism of allosteric antagonism by Czon1107 at α3β4 nAChRs was investigated. Thus, we designed single amino acid analogues of Czon1107 and determined the electrophysiological activity on α3β4 nAChRs to investigate this mechanism.
- Figure 2: A side view is proposed to show the potential interaction of the α3β4 receptor with Czon1107. To complement this, it would be interesting to have a top view of the membrane (like in Fig. 1B), extracellular side. This would clearly show the interaction sites of the toxin and nicotine, relative to each subunit.
Response:
A top view of hα3β4 nAChR docked with Czon1107 is now added as Figure 2F.
- In electrophysiology, if the ratio of α3β4 subunits is 1:1, it can be assumed to be expressed in two stoichiometric forms: α32β43 and α33β42. Yet the authors based their analysis on the α32β43 form. How can the levels of cell current inhibition by the toxin be interpreted if both forms of receptor co-exist? Figure 3 is incomplete. It only offers the levels of inhibition of Czon1107 toxin and its variants on ACh-induced current (300 µM) on α3β4 receptor-expressing oocytes. The concentration of ACh is estimated to correspond to the EC50 for the α3β4 receptor but the dose-response curve is not performed in this study (simply proposed as a reference). This curve should be shown in this figure. The authors should also add to this figure representative current traces of each toxin (Czon1107 and its analogues), including a control current with ACh, and the current with ACh + Czon1107 or its analogues.
Response:
Due to the similarity of the α32β43 and α33β42 forms, we can analyze the mechanism of action based on the MD results of α32β43.
The ACh EC50 for hα3β4 nAChR used in this study is based on Cuny et al., 2016 (DOI: 10.1074/jbc.M116.730804; ref. no. 34). We have added the citation for this paper in the text and references. We have also included representative traces for Czon1107 and four of its analogues (G1R, F2A, S4R and P7R) as Figure 3A.
- The results are commented but not discussed. Clearly a real discussion is missing in this work, which would allow to confront the proposed data to the already known data on the topic of α-conotoxins and nAChRs.
Response:
We have revised the Conclusion section to “Conopeptides such as AuIB [39], RegIIA [40] and their analogues antagonize α3β4 nAChR in a competitive manner. Some conopeptides including the globular disulfide isomer of AuIB [28], GeXIVA [41] and ImII [42], which have multiple disulfide bonds and complex synthesis pathways, have been suggested as non-competitive nAChR antagonists. Uniquely, Czon1107, a single disulfide bond conopeptide, also inhibits the α3β4 nAChR in a non-competitive manner.
This study gives insight into the structure–activity relationship of Czon1107 with hα3β4 nAChR. Through molecular docking and MD simulations, the favorable binding site for Czon1107 is putatively located at the α3 subunit .MD simulations of the interactions also suggest Czon1107 residue F2 forms hydrophobic interactions with residues P264 and P271 of the α3 subunit. Furthermore, Czon1107 residue R3 interacts with residues S225 and T280 of the receptor via hydrogen bonds. These findings provide an experimental and theoretical basis for the structural optimization and mechanism of action of Czon1107 and its analogues. Based on the current experimental results, multi-site mutation is a promising research idea for finding a potent conopeptide at the α3β4 nAChR.”
Minor
L33: a transmembrane domain (TMD)
Response:
Revised as suggested.
Reviewer 2
- The attribution of Czon1107 peptide to α-conotoxins cannot be considered reasoned. Its single-disulfide cysteine framework does not correspond to α-conotoxins. Belonging to the A-superfamily has not been confirmed because there is no analysis of gene sequence from transcriptomic data. Finally, the revealed affinity for the two nAChR subtypes is quite low (> 10 μM). A much more reasonable conclusion seems to be the discovery of a new type of cysteine-poor conopeptides with an unknown target of action, which showed a low affinity for distinct nAChR subtypes, than the attribution of Czon1107 to α-conotoxins. In this regard, it is worth noting that there are already numerous examples of the action of known ligands on a "second" target with an efficiency significantly higher than the affinity of Czon1107 to nAChR (for example, the action of potassium channel blockers from scorpion venom on the same nAChRs). It is important that the discoverers of Czon1107 themselves (see [Mohan et al JBC 2020]) carefully avoided calling it α-conotoxin (with one exception "for the purpose of the following discussion"), reasonably designating it a conopeptide. Taking into account all of the above, I strongly recommend at the moment to transfer Czon1107 from α-conotoxin to conopeptide.
Response:
We have now referred to Czon1107 as a conopeptide.
- The synthesized set of new single-mutated alanine and arginine analogues of Czon1107, unfortunately, did not lead to the discovery of a compound with an essentially changed activity to the receptor, although some of them showed a significantly reduced/improved affinity at the level of 10-20%. Taking into account the initially very low affinity of the wild-type compound (15 μM), it seems rather problematic to draw any conclusions on the structural organization of the binding site (especially allosteric one, little characterized earlier). It is clear that an additional series of mutants is required here, including compounds with multiple mutations, but this is the future work. According to the data presented, the work clearly lacks experimental studies. To smooth out this impression, I recommend supplementing Figure 3 with the raw traces of Czon1107 and some its analogues from electrophysiological measurements.
Response:
Although the Czon1107 analogues showed minimal changes in activity, we can determine whether the side chains of related amino acids play a role in the interactions based on these results. We also propose to carry out multiple site mutations based on the current results in future work to find peptides with enhanced activity. We have also included representative traces for Czon1107 and four of its analogues (G1R, F2A, S4R and P7R) as Figure 3A.
- The computer part of the study offers us the allosteric binding site of Czon1107 on hα3β4 nAChR and even indicates possible atomic bonds between the amino acid residues of the α3 subunit and the peptide. That would be a remarkable scientific advance. However, given the low affinity of the peptide itself and not very noticeable changes in the affinity of its analogues, the presence of this allosteric site requires confirmation by appropriate mutagenesis of the receptor, but for now it remains very hypothetical, that implies a softening of the formulation of the final conclusions. There is also some confusion in the manuscript with the numbering of amino acid residues in the sequence of the α3 subunit. The N-terminal ECDs of all nAChR subunits consist of approximately 200-220 amino acid residues, so the amino acid numbers in TMD should start from this numbers. What are these residues - P(34, 88, 85), T(38, 86, 104), L(76, 101), S49, I83 – attributed to TMD in Figure 4? Where is the numbering from? If you think this numbering is correct, then give in one of the figures the sequence of α3 subunits with this numbering.
Response:
Although we did not perform mutagenesis of the receptor, based on our model, we could explain the selectivity of Czon1107 for hα3β4 versus hα7. Residue F2 is more likely to form hydrophobic interaction with residue P264 of hα3β4, compared to the corresponding hydrophilic and charged D266 of hα7. We have revised Section 2.3.
We have numbered the sequences of α3β4 nAChR as originally reported.
Minor remarks:
- Line 32 – not sigma-, but gamma-subunit.
Response:
Revised as suggested
- Line 34 – ring (??) A, B,… Do you mean loops A, B,…?
Response:
Revised as suggested
- Lines 50 and 67 – reference [14] is out of place here.
Response:
Both errors have been corrected. The correct reference for line 50 is Terlau and Olivera, 2004 [ref. no. 14] and for line 67, the correct reference is Mohan et al., 2020 [ref. no. 30].
- Lines 76-77 – please, specify the amino acid residues of which receptor subunit are indicated.
Response:
Czon1107 interacts with the α3 subunit of α3β4 nAChR. We have revised the lines 76-77 to “Overall, the inhibitory mechanism of Czon1107 at hα3β4 nAChR putatively involves hydrophobic interactions between Czon1107 residue F2 with α3 residues P264 and P271, and hydrogen bonds between Czon1107 residue R3 and residues S225 and T280 of the α3 subunit.”
- Line 112 – reference [34] is identical to reference [30]. Please, correct the list of references.
Response:
Revised as suggested
- Lines 228-229 – For this kind of electrophysiological research, it does not seem very legitimate to use the Student's test in statistical analysis. It would be more correct to use ANOVA and Post-hoc test.
Response:
The Student’s T-test is used appropriately to compare between the means of two groups (e.g. wild-type versus mutant peptide).
Round 2
Reviewer 1 Report
The article "Single-disulfide Conopeptide α-Conotoxin Czon1107, an Allosteric Antagonist of the Human α3β4 Nicotinic Acetylcholine Receptor" by Ma et al. has been revised with substantial corrections.
Please find below what should be modified:
- Figure 2F is too small and the toxin Czon1107 cannot be seen. As it stands, it is not clear. This figure should be exactly like Figure 1B with the 5 subunits docked with the toxin and at the same size. Please remove all the membrane lipids around the receptor.
- Figure 3 has been competed with current traces from a3b4-expressing oocytes. I suggest showing a current trace for every toxin analogue, even in a supplemental figure. Did you test all peptides on the same oocytes? Dis you check for the ACh-mediated current recovery after the peptides?
Minor:
L387: “2016, 291, 23779–23792Zhangsun, D.; . Chem” to be corrected
Author Response
Reviewer 1
1.Figure 2F is too small and the toxin Czon1107 cannot be seen. As it stands, it is not clear. This figure should be exactly like Figure 1B with the 5 subunits docked with the toxin and at the same size. Please remove all the membrane lipids around the receptor.
Response:
A top view of hα3β4 nAChR docked with Czon1107 is now revised as Figure 2F.
- 2. Figure 3 has been competed with current traces from a3b4-expressing oocytes. I suggest showing a current trace for every toxin analogue, even in a supplemental figure. Did you test all peptides on the same oocytes? Dis you check for the ACh mediated current recovery after the peptides?
Response:
Traces for the remaining Czon1107 analogues have been included as Supplemental Figure S30. We have included the following statements “Oocytes were obtained from three frogs” and “Peptides were tested on the same oocyte whenever possible.” to the Materials and Methods section. We have added the following “All peptides inhibited hα3β4-mediated ACh currents in a reversible manner” to the Results section.
Minor
L387: “2016, 291, 23779–23792Zhangsun, D.; . Chem” to be corrected
Response:
Revised as suggested.
Reviewer 2 Report
The corrected version of the manuscript looks noticeably better.
Author Response

(The authors gave the same response as above.)
